# Ovothiol A is the Main Antioxidant in Fish Lens

**DOI:** 10.3390/metabo9050095

**Published:** 2019-05-10

**Authors:** Vadim V. Yanshole, Lyudmila V. Yanshole, Ekaterina A. Zelentsova, Yuri P. Tsentalovich

**Affiliations:** 1International Tomography Center SB RAS, Institutskaya 3a, 630090 Novosibirsk, Russia; vadim.yanshole@tomo.nsc.ru (V.V.Y.); lucy@tomo.nsc.ru (L.V.Y.); zelentsova@tomo.nsc.ru (E.A.Z.); 2Novosibirsk State University, Pirogova 2, 630090 Novosibirsk, Russia

**Keywords:** antioxidant, mass spectrometry, freshwater fish, NMR spectroscopy, ovothiol A

## Abstract

Tissue protection from oxidative stress by antioxidants is of vital importance for cellular metabolism. The lens mostly consists of fiber cells lacking nuclei and organelles, having minimal metabolic activity; therefore, the defense of the lens tissue from the oxidative stress strongly relies on metabolites. Protein-free extracts from lenses and gills of freshwater fish, *Sander lucioperca* and *Rutilus rutilus lacustris*, were subjected to analysis using high-field ^1^H NMR spectroscopy and HPLC with optical and high-resolution mass spectrometric detection. It was found that the eye lenses of freshwater fish contain high concentrations of ovothiol A (OSH), i.e., one of the most powerful antioxidants exciting in nature. OSH was identified and quantified in millimolar concentrations. The concentration of OSH in the lens and gills depends on the fish genus and on the season. A possible mechanism of the reactive oxygen species deactivation in fish lenses is discussed. This work is the first to report on the presence of OSH in vertebrates. The presence of ovothiol in the fish tissue implies that it may be a significantly more common antioxidant in freshwater and marine animals than was previously thought.

## 1. Introduction

Aliphatic and aromatic thiols, being good electron or hydrogen donors and excellent radical scavengers, play an essential role in cellular protection against oxidative stress. The most known thiol, glutathione (GSH), is a major cellular antioxidant in different human and animal tissues [1]. Other aliphatic thiols, including coenzyme M, trypanothione, mycothiol, were found in a variety of animals and microorganisms [2]. The aromatic thiols (thiolhistidines) are not so widespread in nature, although their antioxidative properties may exceed those of aliphatic thiols. In particular, the oxidation potential of ovothiol A (1-methyl-4-thiol-L-histidine, OSH, Scheme 1) at neutral pH has been reported to be significantly lower than that of GSH [3]; the oxidation of OSH by electron acceptors (Fremy’s salt, ferricytochrome *c*, H_2_O_2_) and the reaction with iodoacetamide (IAM) proceed with the higher rate constants than that for GSH [4,5]. A distinct feature of OSH is a very low pKa value of the thiol group (pKa ≈ 1.0–1.4) [4,6,7,8]; therefore, under physiological conditions, it exists predominantly in the more reactive thiolate form. Apparently, this feature is responsible for the high nucleophilicity and one-electron donor ability of OSH [5,6,9,10,11,12,13,14]. So far, OSH and its methylated derivatives were found in the eggs and ovarian tissue of marine invertebrates (such as sea urchin, sea star, scallop, octopus) [15,16,17,18,19]. It has been proposed that the major role of OSH is the cell protection against H_2_O_2_ produced in a respiratory burst in the early part of embryonic activation. OSH was also found in annelids [20], salmon eggs [21], trypanosomatids [5]. In the present work, we report the finding of high concentrations of OSH in the eye lenses of freshwater fish, pike-perch (*Sander lucioperca*) and Siberian roach (*Rutilus rutilus lacustris*).

## 2. Results

Protein-free extracts from the fish lenses were subjected to NMR and high-resolution LC-MS analysis. LC-MS data revealed a strong distinct LC peak with the major MS signals m/z 401.1058, 202.0643 and 127.0325 corresponding to the oxidized form of ovothiol (OSSO, Scheme 1) and its fragments formed during the electrospray ionization.

Further analysis of LC-MS data showed a covalent binding of OSH molecule to other thiols, OSH-Cys (m/z 321.0681 [M+H^+^], m/z 202.0643 [OSH fragment], and m/z 120.0112 [Cys fragment]) and OSH-GSH (m/z 254.0696 [M+2H^+^], m/z 507.1326 [M+H^+^], and m/z 202.0643 [OSH fragment], m/z 308.0904 [GSH fragment]). It is important to note that OSH is readily oxidized in air yielding disulfides, and the formation of oxidized thiols could occur during the sample preparation and chromatographic separation.

Two signals in the NMR spectrum of the lens extract (doublet at 3.71 ppm and quartet at 8.23 ppm with the same constants of spin-spin coupling J=0.5 Hz, Figure 1) were tentatively attributed to NCH_3_ and imidazole CH protons of ovothiol [7,22]. We made an attempt to determine which form of ovothiol (OSH or OSSO) is initially present in the fish lens, and performed several control experiments, monitoring the positions of ovothiol CH (8.23 ppm) and CH_3_ (3.71 ppm) signals in the NMR spectrum.
(A)During the lens homogenization, 57 mM IAM was added into the homogenizing solution to protect thiol SH groups from oxidation. The metabolites were extracted, lyophilized, and dissolved in deuterated phosphate buffer (pH 7.2, 20 mM), and then the NMR spectrum was obtained. It was found that the signal of CH group shifted to 7.68 ppm, and the signal of CH_3_ group—to 3.65 ppm. Since IAM readily reacts with reduced thiols, one can conclude that ovothiol in the lens is present in the reduced form, and the observed signals correspond to carbamidomethylated ovothiol.(B)3 mM dithiothreitol (DTT) was added into NMR tube containing the lens extract in deuterated buffer. The sample was incubated for 12 h at room temperature; then, the NMR spectrum was obtained. No changes in the positions or intensities of the signals from CH and CH_3_ groups were found. DTT reduces S-S bonds; therefore, the obtained result confirms that the concentration of oxidized ovothiol OSSO in the lens extract is much lower than that of OSH.(C)10 mM H_2_O_2_ was added into NMR tube containing the lens extract in deuterated buffer. After incubation for 1 h at room temperature, the NMR spectrum was obtained. The signal of CH group shifted to 7.75 ppm, and the signal of CH_3_ group shifted to 3.65 ppm. Then, the excess of DTT was added into the same sample and incubated for 12 h. After that treatment, the ovothiol signals in the NMR spectrum returned to their starting positions of 8.23 ppm (CH group) and 3.71 ppm (CH_3_ group). This experiment unambiguously demonstrates that ovothiol in the lens extract is present in the reduced form, which undergoes oxidation in the presence of hydroperoxide, and turns back into reduced form in the presence of DTT.

To characterize ovothiol, the extract was subjected to the tandem LC-MS/MS analysis. Besides, the ovothiol-containing fraction was separated and collected using the same LC-MS setup, and then analyzed by ^1^H NMR. We found that the collected fraction contains ovothiol in the oxidized form OSSO (Figure 1); that means that during the LC separation, OSH underwent oxidation. To record the NMR spectra of both OSH and OSSO, the NMR measurements of the fraction were performed with and without DTT addition, respectively (Figure 1). The following data have been obtained:

OSH:

^1^H NMR (700 MHz, D_2_O, 25 °C, DSS): 3.210 (dd, J = 6.9, 16.0 Hz, 1H; βH); 3.255 (dd, J = 5.4, 16.0 Hz, 1H; βH); 3.707 (d, J = 0.5 Hz, 3H; NCH_3_); 4.083 (dd, J = 5.4, 6.9 Hz, 1H; αH); 8.228 (q, J = 0.5 Hz, 1H; Im 2H).

MS/MS (ESI+, 35 eV): 202.0643 [C_7_H_12_N_3_O_2_S^+^], 185.0377 [C_7_H_9_N_2_O_2_S^+^], 167.0272 [C_7_H_7_N_2_OS^+^], 156.0589 [C_6_H_10_N_3_S^+^], 141.0480 [C_6_H_9_N_2_S^+^], 127.0325 [C_5_H_7_N_2_S^+^], 114.0245 [C_5_H_7_N_2_S^+^], 100.0215 [C_4_H_6_NS^+^], 86.0059 [C_3_H_4_NS^+^].

OSSO:

^1^H NMR (700 MHz, D_2_O, 25 °C, DSS): 2.653 (dd, J = 8.0, 15.6 Hz, 1H; βH); 2.836 (dd, J = 6.9, 15.6 Hz, 1H; βH); 3.653 (s, 3H; NCH_3_); 3.668 (dd, J = 6.9, 8.0 Hz, 1H; αH); 7.750 (s, 1H; Im 2H).

MS/MS (ESI+, 35 eV): 401.1061 [C_14_H_21_N_6_O_4_S_2_^+^], 356.0845 [C_13_H_18_N_5_O_3_S_2_^+^], 234.0365 [C_7_H_12_N_3_O_2_S_2_^+^], 202.0646 [C_7_H_12_N_3_O_2_S^+^], 201.0567 [C_7_H_11_N_3_O_2_S^+^], 200.0488 [C_7_H_10_N_3_O_2_S^+^], 185.0377 [C_7_H_9_N_2_O_2_S^+^], 170.0924 [C_7_H_12_N_3_O_2_^+^], 168.0766 [C_7_H_10_N_3_O_2_^+^], 156.0589 [C_6_H_10_N_3_S^+^], 127.0325 [C_5_H_7_N_2_S^+^], 114.0245 [C_5_H_7_N_2_S^+^].

More detailed data on ovothiol characterization are presented in Appendix A, including NMR and MS/MS spectra. The obtained results are in a good agreement with the previously reported data [7,15,22] on ovothiol NMR and MS properties, and confirm the presence of ovothiol in the extract from the fish lens.

The measurements of lenticular OSH concentrations were performed for *S. lucioperca* and *R. rutilus lacustris* caught in the Ob reservoir (Novosibirsk region, Siberia, Russia) during the late autumn (October-November) and winter (February) periods. The exact dates of catching are given in Appendix A. The concentrations were determined from the integration of signals in the NMR spectrum (Figure 1) relatively to an internal standard sodium 4,4-dimethyl-4-silapentane-1-sulfonic acid (DSS) present in all samples in the concentration of 6 × 10^−6^ M. The results are presented in Table 1, together with the data on histidine (OSH precursor) and GSH concentrations measured in the same samples.

The variations of concentrations within each group are relatively small, and typical standard deviation for most of measurements can be estimated at 15–20%. At the same time, the differences between genera and between species belonging to the same genus but caught at different seasons are rather significant. In particular, the level of OSH in lenses of *S. lucioperca* decreases from 3 µmol/g in autumn to 1.6 µmol/g in winter (1.9-fold change), and in lenses of *R. rutilus lacustris* from 1.1 µmol/g to 0.27 µmol/g (a 4.1-fold change). Most likely, this difference should be attributed to the low oxygen level in water and low activity of fish during the winter. Typically, ice freezes in Novosibirsk region in the second half of November, while the ice-breaking occurs at the end of April. Low oxygen level in water during the winter leads to the deceleration of metabolic processes in fish, and, probably, to the slower synthesis of OSH in the fish lens. However, the levels of another lens antioxidant GSH (which is also synthesized inside the lens) [23] change less significantly: in lenses of *S. lucioperca* the GSH levels in autumn and in winter are the same (Table 1), while in *R. rutilus lacustris* lenses, only a 1.8 times decrease was observed. Another possible explanation of the low OSH level in the fish lens during the winter might be an insufficient histidine supply due to the low feeding activity. The drop in the histidine level in the fish lenses in the late winter (Table 1) supports this assumption.

OSH was also detected in gills of both *S. lucioperca* and *R. rutilus lacustris* caught in winter, but its level in the gills is significantly lower than that in the lens: 100 ± 90 nmol/g (*R. rutilus lacustris*) and 270 ± 80 nmol/g (*S. lucioperca*).

Finding of OSH in millimolar concentrations in the fish lens changes the modern conception of redox processes occurring in the lens. The concentration of OSH in the fish lens is much higher than that of GSH, and the rates of OSH reactions with reactive oxygen species (ROS) are higher [3,6,10,24]. That means that in the fish lens, OSH represents the first line of the cellular defense against oxidative stress. At the same time, it is known that the equilibrium in the Equation (1)
2 GSH + OSSO ↔ 2 OSH + GSSG(1)
is shifted well to the right [2,6]. Therefore, the general scheme of the ROS deactivation by OSH-GSH system in the fish lens can be presented as following: (1) OSH reduces ROS forming OSSO; (2) GSH reduces OSSO forming GSSG; (3) glutathione reductase reduces GSSG.

## 3. Discussion

Tissue protection using the OSH-GSH system might be used in a therapeutic practice by the application of ovothiol as an additional antioxidant to tissues with the natural intracellular GSH synthesis and reduction. In particular, ovothiol can be used as an anti-inflammatory drug [25]. It is likely that, at the moment, the therapeutic potential of ovothiol is largely underappreciated [14].

In several papers [26,27,28,29,30] published during the last decade, it has been reported that the low dietary histidine supplementation provokes the cataractogenesis in farmed salmon. This effect was attributed to the insufficient synthesis of the major osmolyte of the fish lens, N-acetylhistidine [31,32], which results in the development of osmotic cataracts after transferring to seawater. Very possible that the low histidine supply also causes a drop in the OSH level in the salmon lens, which makes the lens more vulnerable to the oxidative stress. That can make a substantial contribution to the cataract development. To check this assumption, it would be interesting to measure the OSH levels in lenses of sea fish, of farmed fish in particular.

Unlike other tissues, the lens has very specific structure, mostly consisting of fiber cells without inner cellular apparatus (nucleus, organelles). Therefore, the defense of the lens tissue from the oxidative stress strongly relies on metabolites. For example, the protection of the human lens is provided, to a large extent, by glutathione and ascorbate. Finding more effective antioxidant than GSH in the fish lens could point at the suitability of OSH for particular living conditions such as large pressure range or seasonal dependence on feeding. Another possible explanation of the absence of OSH in tissues of terrestrial animals is that the pathway responsible for the OSH synthesis [14] was lost during the evolutionary mutations in a similar way as it happened to another vital antioxidant, ascorbic acid [33]. Biosynthesis of OSH requires the coupling of histidine and cysteine molecules, this reaction is catalyzed by enzyme OvoA [14,34,35]. This enzyme was first characterized using recombinant OvoA from protistic *T. cruzi* and *E. tasmaniensis* [35]. In the recent paper of Castellano et al. [34] it was suggested that the gene coding OvoA was lost in bony fishes. This suggestion was made on a basis of evolutionary analysis of OvoA in metazoans: OvoA orthologous were not identified in bony vertebrates. Observation of high concentrations of OSH in the fish lens poorly reconcile with the suggestion about the OvoA loss in bony fishes. This discrepancy can be attributed to either low quality of the genome assembly present in the databases, which could result in missing OvoA during the database search, or to an alternative pathway of the OSH biosynthesis without participating of OvoA operating in bony fishes. Another possible explanation is that OSH in the fish tissues has an exogenous origin, and a fish acquires OSH with food (such as freshwater crustaceans or clams inhabiting the Ob River basin and serving as an important source of nutrition for *R. rutilus lacustris*), and accumulates OSH in the lens using OSH-specific transporter proteins. However, this hypothesis does not explain which way OSH is acquired by predatory fishes, and why the concentrations of OSH in the *S. lucioperca* (predator) lens and gills is significantly higher than that in the *R. rutilus lacustris* lens and gills. It is also possible that OSH is produced by symbiotes living inside the fish. Altogether, the questions of the OSH origin and of the mechanism leading to its concentrating in the fish lens remain open.

OSH is highly unstable compound; for example, after several days of the extract storage in a capped NMR tube (aqueous solution, in darkness, +4 °C) the OSH signal disappears completely. Moreover, significant OSH oxidation was found to occur during the chromatographic separation. That means that the detection and quantification of ovothiol requires special caution, and it could easily be overlooked in previous metabolomic studies. In particular, in the present work, ovothiol was also detected in gills of both *S. lucioperca* and *R. rutilus lacustris*, although in significantly lower concentrations than that in lenses. At the same time, in the recent studies of metabolomic composition of fish tissues (lens, gills, kidney, liver) [30,36,37] the presence of ovothiol was not reported. Therefore, ovothiol may be significantly more common metabolite in freshwater and marine animals than it was thought, and it probably plays an important role in the antioxidant cell protection in various tissues and species.

## 4. Materials and Methods

### 4.1. Sample Preparation

The animals were treated according to the ARVO Statement for the Use of Animals in Ophthalmic and Vision Research and the European Union Directive 2010/63/EU on the protection of animals used for scientific purposes, with the ethics clearance from the International Tomography Center SB RAS. Pike-perch (*Sander lucioperca*, body weight 200–300 g) and Siberian roach (*Rutilus rutilus lacustris*, body weight 80–110 g) were caught in the Ob reservoir: *S. lucioperca*—in October (n = 8) and February (n = 7); *R. rutilus lacustris*—in November (n = 10) and February (n = 5). The fish were killed with a concussive blow to the head immediately after catching; subsequently, the lenses and gills were removed, frozen and kept at −70 °C until analyzed.

Each fish lens was weighed prior to homogenization; for *S. lucioperca*, the typical lens weight was 100 mg, and for *R. rutilus lacustris* −40 mg. Only one lens from each fish was used for the analysis. The lens was placed in a glass vial and homogenized with a TissueRuptor II homogenizer (Qiagen, Venlo, The Netherlands) in 1600 μL of cold (−20 °C) MeOH, and then 800 μL of water and 1600 μL of chloroform was added. The mixture was shaken in a shaker for 20 min and left at −20 °C for 30 min. Then the mixture was centrifuged at 16100× *g*, + 4 °C for 30 min, and the upper (MeOH-H_2_O) layer was collected. The collected supernatant was divided into two parts for NMR (2/3) and LC-MS (1/3) analyses. The fish gills were treated in the same way.

The extracts for NMR measurements were lyophilized, re-dissolved in 600 μL of D_2_O containing 6 × 10^−6^ M sodium 4,4-dimethyl-4-silapentane-1-sulfonic acid (DSS) as an internal standard and 20 mM deuterated phosphate buffer to maintain pH 7.2. The extracts for LC-MS analysis were lyophilized, re-dissolved in 100 μL of aqueous solution containing 10 μM N-acetyltryptophanamide as an internal control.

### 4.2. NMR Measurements

The ^1^H NMR measurements were carried out with the use of a NMR spectrometer AVANCE III HD 700 MHz (Bruker BioSpin, Rheinstetten, Germany) equipped with a 16.44 Tesla Ascend cryomagnet. The proton NMR spectra for each sample were obtained with 96 accumulations. The temperature of the sample during the data acquisition was kept at 25 °C, and the detection pulse was 90 degree. The repetition time between scans was 20 s to allow for the relaxation of all spins. Low power radiation at the water resonance frequency was applied prior to acquisition to presaturate the water signal. The concentrations of metabolites in the samples were determined by the peak area integration respectively to the internal standard DSS.

### 4.3. LC-MS Measurements

The LC separation was performed on an UltiMate 3000RS chromatograph (Dionex, Germering, Germany) using a hydrophilic interaction liquid chromatography (HILIC) method on a TSKgel Amide-80 HR (Tosoh Bioscience, Griesheim, Germany) column (4.6 × 250 mm, 5 μm). The chromatograph was equipped with a flow cell diode array UV-vis detector (DAD) with 190–800 nm spectral range. Solvent A consisted of 0.1% formic acid solution in H_2_O, solvent B consisted of 0.1% formic acid solution in acetonitrile. The gradient was (solvent B): 95% (0–5 min), 95–65% (5–32 min), 65–35% (32–40 min), 35% (40–48 min), 35–95% (48–50 min), 95% (50–60 min); the flow rate was 1 mL/min, the sample injection volume was 10 μL, the column temperature was 40 °C. After the DAD cell a home-made flow splitter (1:10) directed the lesser flow to an ESI-q-TOF high-resolution hybrid mass spectrometer maXis 4G (Bruker Daltonics, Bremen, Germany). The mass spectra were recorded in a positive mode with 50–1000 m/z range. The MS setup, the calibration procedure, and the data processing were described in details earlier [38]. The LC-MS/MS measurements were performed in AutoMS/MS mode. During the LC fraction collection, the sample injection volume was 30 μL, and the fraction content was monitored by MS signal.

### 4.4. Statistics

The data were analyzed using the statistical package Statistica 6.0 (StatSoft, Tulsa, OK, USA). The sample size for each experimental group is given in Table 1. The data in Table 1 represent mean value ± standard deviation.

### 4.5. Data Availability

NMR raw data obtained in this study have been deposited in MetaboLights repository, study identifier MTBLS806 (https://www.ebi.ac.uk/metabolights/MTBLS806). All data generated and analyzed are reported in this article, including ovothiol MS/MS data (Appendix A) and concentrations of ovothiol, histidine and GSH in individual fish lenses (Appendix A).

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
