# Peer review of "Ovothiol A is the Main Antioxidant in Fish Lens"

_metabolites, 2019, doi:10.3390/metabo9050095_

Round 1

Reviewer 1 Report

General remarks

The authors firstly identified ovothiol, one of antioxidants, from teleost fish. Overall, the manuscript is well written and contained valuable data. However, the manuscript is needed to be reconstructed and requires additional explanation. If the authors make substantial changes to the manuscript and make necessary amendments, the reviewer suggests that the revised article is suitable for publication in metabolite

Major comments

Generally, it is well known that levels of antioxidants is enhanced by oxidative stress. Therefore many literatures include the alteration of antioxidant levels in organism after exposure to ambient stress, such as anthropogenic harmful chemicals. In the present study, however, the authors did not expose fish to such ambient stress. Did they measure baseline levels of ovothiol? They should make it clearer. 

Minor comments

Introduction

Since the authors measured levels of histidine as well as OSH and GSH in the present study, they should explain why they need to measure histidine.

Result

Lines 54-122: the section should be moved to materials and methods.

Lines 105-122: The authors should explain NMR spectra and MS/MS data about histidine.

Line 135: For GSH concentrations in R. rutilus captured in November, standard deviation is higher than 20% (280 ±130).

Line 142: Even though the deceleration of metabolic processes of fish in winter is low, OSH level of R. rutilus in November (1070) is lower than that of S. lucioperca in February (1630). They should explain why.

Discussion

Line 197: The authors should indicate the result of gills.

Lines 208-209: Do the authors mean previous reports (refer no. 30,36,37) aimed to identify OSH, but they unfortunately could not determine in their biological samples due to inappropriate sample pretreatment? Or don’t they target OSH? Please make it clear since it is important to highlight your results. If the former is correct, what is the key technique in identification of OSH from the sample?

Table1 and 4.4 statistics

The authors should indicate the results of OSH concentrations of gills, even though they were not significant since the authors mentioned in line 205-207. They also must show indicate the results of OSSO.

The authors should carried out statistical analysis and indicate analytical outcomes.

Author Response

Dear Editor,

We thank reviewers for helpful comments and suggestions. Below are our point-to-point replies.

Reviewer 1.

Comment: Generally, it is well known that levels of antioxidants is enhanced by oxidative stress. Therefore many literatures include the alteration of antioxidant levels in organism after exposure to ambient stress, such as anthropogenic harmful chemicals. In the present study, however, the authors did not expose fish to such ambient stress. Did they measure baseline levels of ovothiol? They should make it clearer.

Reply: That is a very good comment; it raises the question on metabolic response to the oxidative stress, and on ovothiol level in the fish tissues under exposure to stress in particular. In fact, the work in this direction is currently in progress in our lab. However, the present paper is just a first short report on the presence of high concentrations of OSH in the fish lens. Apparently, there are many questions to be answered, including OSH distribution among different fish tissues, the origin of OSH in the fish tissues, and the influence of environmental factors on OSH concentration. Answers to all these questions, including the baseline levels in different types of fish, are beyond the scope of this work.

Comment: Since the authors measured levels of histidine as well as OSH and GSH in the present study, they should explain why they need to measure histidine.

Reply: Histidine is the precursor in OSH biosynthesis, now that is mentioned in the manuscript (line 129).

Comment: Lines 54-122: the section should be moved to materials and methods.

Reply: We do not agree with this comment. Ovothiol identification, validation, and also determination of its state (reduced or oxidized form) in the fish lens are the important results of the work.

Comment: Lines 105-122: The authors should explain NMR spectra and MS/MS data about histidine.

Reply: NMR and MS/MS data in lines 105-122 correspond to reduced (OSH) and oxidized (OSSO) ovothiol. The data are given in commonly accepted form. Additional information and signal assignment are presented in Supplementary material. Histidine is a well-known compound, and its NMR and MS/MS data are presented in databases.

Comment: Line 135: For GSH concentrations in R. rutilus captured in November, standard deviation is higher than 20% (280 Â±130).

Reply: We agree. The text in line 136 is corrected.

Comment: Even though the deceleration of metabolic processes of fish in winter is low, OSH level of R. rutilus in November (1070) is lower than that of S. lucioperca in February (1630). They should explain why.

Reply: OSH level in lenses of S. lucioperca is always significantly higher than in lenses of R. rutilus lacustris, and seasonal variations are not able to compensate this difference. We do not have a good explanation for the reason of this difference.

Comment: Line 197: The authors should indicate the result of gills.

Comment: Table 1: The authors should indicate the results of OSH concentrations of gills, even though they were not significant since the authors mentioned in line 205-207. They also must show indicate the results of OSSO.

Reply: The measured levels of OSH in the fish gills are now given in Lines 151-152. OSSO was not detected in any sample. That is not surprising since all samples contained reduced glutathione GSH which rapidly reduces oxidized ovothiol (Reaction (1) in the manuscript).

Comment: Lines 208-209: Do the authors mean previous reports (refer no. 30,36,37) aimed to identify OSH, but they unfortunately could not determine in their biological samples due to inappropriate sample pretreatment? Or don’t they target OSH? Please make it clear since it is important to highlight your results. If the former is correct, what is the key technique in identification of OSH from the sample?

Reply: Previous reports mentioned in the manuscript (refer no. 30,36,37) correspond to NMR-based untargeted metabolomics, and they were not aimed on OSH identification. It is difficult to judge why OSH was not reported in these works. The possible reasons include: 1) OSH underwent oxidation during the sample preparation; 2) OSH was present in samples, but was not identified, and, therefore, was not reported; 3) probably (although unlikely) OSH is present in tissues of freshwater fish, but it is absent in marine fish. Since we do not have arguments in favor of any of these hypotheses, we prefer to avoid speculations on the matter.

Comment: The authors should carried out statistical analysis and indicate analytical outcomes.

Reply: The present work includes the analysis of content of only three compounds in the fish lens (histidine, OSH, and GSH), and, in our opinion, the presented mean values and standard deviations are sufficient for this type of works. Supplementary material also includes the individual data for every fish taken into study. The full-scale statistical analysis is more appropriate for metabolomic study involving tens or hundreds of metabolites; the paper on the metabolomic composition of the fish lens and gills is currently under preparation in our lab, and it will include the correct statistical analysis.

Reviewer 2 Report

This manuscript is to analyze the antioxidant in fish lens by NMR, and the results clearly showed that the ovothiol A is the main antioxidant in the fish including Sander lucioperca and Rutilus rutilus lacustris. This manuscript is interesting and reasonable. Therefore, I do sincerely suggest it to be accepted in your esteemed journal after a minor revision.

1.      The discussion on the difference in histidine, OSH and GSH of fish in different sampling times is inappropriate. The authors considered that the low oxygen level and low activity of fish during the winter might be the main reason of lower antioxidants in fish. Why not other parameters? How they know the dissolved oxygen (DO) is lower in winter? Did they detect the DO in water? A rigorous discussion is needed.

2.      The information about fish sampling is unclear. The sampling times, October and February? In the same year or different?

3.      The listed references should be readdressed to meet the journal style and should be identical from the beginning to the end.

Author Response

Dear Editor,

We thank reviewers for helpful comments and suggestions. Below are our point-to-point replies.

Reviewer 2.

Comment: The discussion on the difference in histidine, OSH and GSH of fish in different sampling times is inappropriate. The authors considered that the low oxygen level and low activity of fish during the winter might be the main reason of lower antioxidants in fish. Why not other parameters? How they know the dissolved oxygen (DO) is lower in winter? Did they detect the DO in water? A rigorous discussion is needed.

Reply: It is well known that DO level in ice-covered lakes is maximal during the late autumn (before ice freezing), and minimal in winter due to the water isolation from atmosphere (for example, see A. Terzhevik and S. Golosov, Dissolved Oxygen in Ice-Covered Lakes, in Encyclopedia of Lakes and Reservoirs, DOI: https://doi.org/10.1007/978-1-4020-4410-6_225). In Ob reservoir, the DO level varies from 9-12 mg/L in autumn to as low as 5 mg/L in winter. Unfortunately, these data are present only in Russian websites (https://water-rf.ru), so we did not include them into discussion. In general, although we believe that the low oxygen level and low activity of fish during the winter are the main reasons of the OSH decrease in winter, other factors also cannot be excluded. Therefore, we agree with the reviewer’s comment, and have softened the sentence formulation (Line 140).

Comment: The information about fish sampling is unclear. The sampling times, October and February? In the same year or different?

Reply: The exact dates of catching are given in Supplementary material. Now that is clearly indicated in the paper (Lines 125-126).

Comment: The listed references should be readdressed to meet the journal style and should be identical from the beginning to the end.

Reply: The reference list is corrected.